# Shifting employment and perceptions of household responsibilities during early stages of the COVID-19 pandemic in Nevada, USA

**Courtney Coughenour** ⬤*, **Lung-Chang Chien, Brian Labus** ⬤, **Maxim Gakh, Pashtana Usufzy**

School of Public Health, University of Nevada Las Vegas, Las Vegas, NV, United States of America

* courtney.coughenour@unlv.edu

## Abstract

### Background

Employment and household responsibility are critical health determinants. The COVID-19 pandemic altered the work and social landscapes in Nevada, USA through closures of workplaces and schools/childcare centers, changing patterns of employment, and household responsibilities. This study aimed to measure changes in employment status and perceived housework responsibilities among Nevada adults in December 2020, before widespread availability of COVID-19 vaccines in a pandemic-affected economy.

### Methods

Using a cross-sectional telephone survey of 1,000 Nevada adults, this study compared respondent experiences and perceptions of employment and time spent on housework in December 2020 to pre-pandemic using multinominal logistic, proportional odds, and logistic models.

### Results

70.52% of participants experienced no employment change; roughly 24% reported being fired/laid-off, working reduced hours, or quitting. Chi-square analyses found participants of color more likely than Whites to report being fired/laid-off or working reduced hours (p-value = 0.0005), though these findings were not significant in our models. Participants in the lowest income bracket had higher odds of being fired/laid off (p-value = 0.0030), and participants aged 65+ were less likely to experience employment change (p-value<0.0001). 32.43% of participants reported more time spent on housework. Multivariate analyses showed age is significantly associated with changes in both employment status (p-value<0.0001) and housework time (p-value<0.0001); income is a significant factor for employment status change (p-value = 0.0024). Of those reporting their households spent more time on housework, females were 2.90 times (95% CI = 1.66, 5.05, p-value = 0.0002) more likely to report taking on the additional work.

**Data Availability Statement:** The data for this paper can be found on Open Science Framework at the following link: https://osf.io/dnufk/.

**Funding:** The publication fees for this article were supported by the UNLV University Libraries Open Article Fund.

**Competing interests:** The authors have declared that no competing interests exist.

## Conclusions

Results demonstrate a disproportionate burden in employment change and/or more perceived household responsibilities on lower income and younger respondents, and females reporting they are more likely to take on additional housework. Understanding systemic vulnerabilities related to employment status and household responsibilities is necessary to aid impacted communities and to plan for future emergencies.

## Introduction

The COVID-19 pandemic transformed work and home landscapes. As the virus spread in early 2020, governments issued emergency declarations [1]. Local, state, and federal prevention-focused policies, such as mask mandates, stay-at-home orders, business and school closures, eviction moratoria, and other precautions were adopted to slow the spread [2]. Individuals' behaviors also changed, with some avoiding businesses over infection fears [3]. By December 2022, the pandemic produced over 6.6 million cumulative hospitalizations and over 1.1 million deaths related to COVID-19 in the U.S. [4].

The pandemic also altered the economy, work, families, and communities [5, 6]. A notable effect was on national unemployment, which reached 14.7% in April 2020, the highest since monthly tracking began in January 1948 [7, 8]. Impacts on jobs were especially stark in certain parts of the U.S., including in Nevada, due to a lack of economic diversification and an economy heavily tied to gaming and tourism, especially around Las Vegas [9–11]. In March 2020, Nevada first issued an emergency order, then a temporary closure of all gaming establishments, and after that a mandatory stay-at-home order [12]. Nevada's economy soon felt the effects of COVID-19 and mitigation efforts. The seasonally adjusted unemployment rate rose to 28.5% in April 2020 amidst the closures, a historic high for Nevada and any U.S. state [13]. Though these historically high rates fell, elevated unemployment in Nevada persisted. Nevada's 9.2% unemployment rate in December 2020, when our survey was conducted, was the second-highest of any U.S. state [14]. Before the pandemic began, unemployment was 3.7% in Nevada and 3.6% nationally [14].

Nevada relaxed some closures in late-spring 2020 [15]. By early October, case rates increased and limitations on gathering were re-applied [12]. By mid-December, Nevada again reported high infection levels, ranking among the top eight states for COVID-19 cases per 100,000 residents [16]; Nevada had over 39 new weekly COVID-19 hospital admissions and 94 cumulative deaths per 100,000 residents [4]. Although initially intended to last a few weeks, the shutdown continued to impact Nevadans through 2020 and beyond.

COVID-19 closures and slowdowns impacted certain residents and households disproportionately in the U.S.. One survey revealed that U.S. seniors were more likely to have used up most or all of their savings or lost their incomes/jobs during the pandemic compared to seniors in other high-income countries [17]. The overall pandemic-related economic hardship rate among U.S. seniors was 19%, but rates were 14% for White, 39% for Latinx/Hispanic, and 32% for Black seniors [17]. Additionally, rural labor markets appear to have performed better than urban markets in terms of the share of adults unable to work or to look for work, though urban-area workers were more likely to report working remotely [18].

Research also indicates a disproportionate burden of school closures on women; mothers across the U.S. were more likely to report cutting back on work hours or leaving the workplace altogether in areas with in-person school or daycare closures [19, 20]. On March 16, 2020,

Nevada closed kindergarten through 12th-grade schools, shifting daytime childcare onto parents and guardians [12]. Prior to the pandemic, women in Nevada consistently experienced lower unemployment rates than men. This shifted after the pandemic began [21].

People of color in Nevada were disproportionately affected by the economic impacts of the pandemic. These communities already faced disproportionate economic burdens, as sectors in which they are significantly represented tend to report wages that are lower than the state average [22]. They also make up about half of the workforce in sectors like retail, accommodation and food services, transportation, healthcare, and social assistance, all of which were affected by pandemic closures. Metropolitan areas with industries most vulnerable to COVID-19 impacts, such as Las Vegas, also tended to contain larger Hispanic/Latinx populations [23]. In April 2020, the national Hispanic/Latinx unemployment rate skyrocketed to a record 18.2% [24].

The pandemic created seismic shifts in employment and household responsibilities across the U.S., disproportionately impacting certain demographics, including people of color and women. Nevada's context may have contributed to especially stark impacts. A better understanding of employment changes and household responsibilities among Nevada adults is necessary and especially important because of Nevada's diverse population mix and economy, which were particularly vulnerable to the COVID-19 pandemic. A better understanding of the challenges created by COVID-19 can help with recovery and mitigation and to plan for future emergencies. This study therefore aimed to measure changes in employment status and perceived housework responsibilities among Nevada adults in December 2020, after the start of the pandemic and prior to the widespread availability and uptake of COVID-19 vaccines in a still-affected Nevada economy.

## Methods

This study utilized a cross-sectional approach to gather and analyze survey responses. The survey was crafted by the study authors based on a review of the literature [17–20] and expert consensus and was conducted by a market research firm. The 1,000 participants were residents of Nevada, a U.S. Mountain West state of roughly 3.1 million people [25]. Nevada ranks third in the national racial and ethnic Diversity Index, with non-Hispanic whites comprising 45.9% of the population [26]. To participate in the study, a respondent had to be: a resident of Nevada, able to be contacted via a phone line, over the age of 18 years, and willing to complete the survey in English. A power calculation based on the chi-square test indicates that this sample size of 1,000 can detect effect sizes between 0.099 and 0.140.

### Data collection and survey

The survey, conducted from December 21 to 28, 2020 and contained 80 questions. Participants were contacted via landline (n = 408) and cellphone (n = 592) between 5 p.m. and 9 p.m. local time by a market research firm who maintains the contact information for participants. Cellphone lines came from a proprietary data source and landlines from directory-listed numbers. Of these sources, phone numbers from individuals residing in Nevada were randomly selected and dialed. Only one participant per household was eligible to participate. Data were anonymized by the research firm prior to analysis. All participants were told the purpose of the research study, the length of time the survey would take to complete, and that their participation was voluntary. All participants were read consent information and verbally consented to participating in the survey by agreeing to the following statement: "I agree to participate in this study. I am at least 18 years of age." All participants completed the survey in English. The Office of Research Integrity at the University of Nevada, Las Vegas reviewed the study protocol

and survey and determined it to be exempt from full IRB review due to the minimal risk of harm and anonymous nature of the survey (December 3, 2020; protocol number 1688442–2).

Participants were asked if they worked for pay before the COVID-19 pandemic, with the options of (1) worked full time (35+ hours/week), (2) worked part-time, or (3) did not work. Though we are unable to ascertain why individuals may not have worked prior to the pandemic, we analyzed both the full sample of participants and a subsample that responded to working either full or part time before the pandemic. Participants were also asked if their employment changed and could report (1) no change, (2) reduced hours, (3) being fired or laid off, (4) quitting work voluntarily, or (5) working more hours since the pandemic began.

Two questions on household responsibilities included: (1) if the amount of time participants or their households spent on housework changed since the start of the pandemic and, (2) if so, who expended more time on housework–the respondent or someone else. Demographic variables collected included age (in years), gender (female; male; transgender; non-binary; choose to self-describe), race/ethnicity (White; Hispanic, Latino(a), or Spanish; Black or African American; Asian; American Indian or Alaska Native; Middle Eastern or North African; Native Hawaiian or other Pacific Islander; some other race or ethnicity [with the option to select all that apply]), educational level (less than high school diploma; high school graduate or GED; some college or technical school; college graduate; graduate or professional degree), and household income (categories ranging from $0 –$120,001 or more). Participants were categorized as living in an urban or rural residential area based on the U.S. Census Bureau's county classification [27].

## Statistical methods

The study applied three modeling approaches to accommodate different types of categorical outcome measures. First, the multinomial logistic model was applied to the 5-level employment status change predicted by the covariates for both the full sample and subsample. The link function of the model is a generalized logit function by contrasting the reference level "has not changed" with each of the other four levels in the employment status change.

Thus, this model contains four separate model equations with four outcomes: $\log\left(\frac{P(reduce\ hours)}{P(has\ not\ changed)}\right)$, $\log\left(\frac{P(fired\ or\ laid\ off)}{P(has\ not\ changed)}\right)$, $\log\left(\frac{P(I\ quit\ working\ voluntarily)}{P(has\ not\ changed)}\right)$, and $\log\left(\frac{P(working\ more\ hours)}{P(has\ not\ changed)}\right)$, sharing the same predictors (gender, age category, race/ethnicity, education, income, and residential area). Second, we applied the proportional odds model to fit the 3-level housework time change. The link function of the proportional odds model is a cumulative logit function by contrasting less housework with more housework. Thus, the proportional odds model consists of two separate model equations for two outcomes in terms of $\log\left(\frac{P(less\ time)}{P(same\ time)+P(more\ time)}\right)$ and $\log\left(\frac{P(less\ time)+P(same\ time)}{P(more\ time)}\right)$, predicted by the same covariates as the multinomial logistic model for employment status change. Third, we applied the logistic model to fit the 2-level perceptions of primary housework responsibility.

In the three models, the estimated coefficient of each predictor was transformed into an odds ratio (OR) by an exponential function. In particular, the OR from the multinomial logistic model explained the odds of a non-reference level (i.e., reduced hours, fired or laid off, quit working voluntarily, and working more hours) versus the odds of the reference level (i.e., no change in employment status.) The OR from the proportional odds model explained the odds of a household spending less time versus the odds of a household spending more time doing housework. The OR from the logistic model explained the odds of the respondent perceiving oneself doing the increased housework versus the odds of perceiving someone else doing the increased housework. This study adopted the complete case analysis, where observations with

a missing value for any dependent or independent variables were removed in the model-fitting, leading to the analysis of 766 responses in the multinomial logistic model, 775 responses in the proportional odds model, and 256 responses in the logistic model.

Data management and analysis were performed in SAS v9.4 (SAS Institute Inc., Cary, North Carolina, USA). The significance level was set to 0.05.

## Results

In total, 1,000 participants completed the survey, with a response rate of 29.9%. Most participants identified as White, followed by Hispanic, and nearly 55% as female. The sample was highly educated, with 38.0% having a 4-year degree or higher and 44.5% reporting a median household income of $65,001 or higher. See Table 1 for the full demographic breakdown. Table 2 shows that 70.52% of the full sample participants did not experience a change in employment status. However, 11.25% of them were fired or laid off. The following covariates

**Table 1. Demographics of a sample of Nevada adults in December 2020 (n = 1000) and state estimates.**

| Category | | Sample | Sample %[d] | Nevada | Nevada % |
|---|---|---|---|---|---|
| Gender[a] (Missing = 9) | | | | | |
| | Female | 544 | 54.89 | 1,558,784 | 49.58 |
| | Male | 447 | 45.11 | 1,585,207 | 50.42 |
| Race & Ethnicity[a] (Missing = 28) | | | | | |
| | Non-Hispanic White | 670 | 68.93 | 1,420,256 | 45.17 |
| | Hispanic, Latino(a), or Spanish | 107 | 11.01 | 940,759 | 29.92 |
| | Non-Hispanic Black or African American | 79 | 8.13 | 274,003 | 8.72 |
| | Non-Hispanic Asian | 56 | 5.76 | 284,759 | 9.06 |
| | Multiple or other races | 60 | 6.17 | 224,214 | 7.13 |
| Age[b] (Missing = 30) | | | | | |
| | 18–29 | 114 | 11.75 | 474,009 | 20.23 |
| | 30–44 | 125 | 12.89 | 621,786 | 26.54 |
| | 45–64 | 293 | 30.21 | 768,953 | 32.82 |
| | 65+ | 438 | 45.15 | 478,020 | 20.40 |
| Education level[c] (Missing = 13) | | | | | |
| | Less than high school diploma | 33 | 3.34 | 273,999 | 13.09 |
| | Grade 12 or GED (high school graduate) | 218 | 22.09 | 584,698 | 27.92 |
| | Some College | 361 | 36.58 | 702,126 | 33.53 |
| | College 4 years or more (college graduate) | 196 | 19.86 | 348,505 | 16.64 |
| | Graduate or professional degree | 179 | 18.14 | 184,492 | 8.81 |
| Income level (Missing = 182) | | | | | |
| | $0-$30,000 | 229 | 28.00 | | Comparable data not available |
| | $30,001-$65,000 | 225 | 27.51 | | |
| | $65,001-$105,000 | 187 | 22.86 | | |
| | $105,001 or more | 177 | 21.64 | | |
| Residential area[a] | | | | | |
| | Urban | 849 | 84.9 | 2,886,098 | 90.95 |
| | Rural | 151 | 15.1 | 287,228 | 9.05 |

a = State estimates are for total population [51, 54].

b = State estimate is for population 18 years of age and older, but percentage estimate is for overall population [52].

c = State estimates are for population ages 25 years of age and older [53].

d = Sample percentage excludes missing cases.

**Table 2. Frequencies and proportions in covariates by employment status change from a sample (n = 987) of Nevada adults in December 2020.** The p-value was computed by the chi-square test.

| Variable | Has not changed (N = 696; % = 70.52) | | Reduced hours (N = 74; % = 7.50) | | Fired or laid off (N = 111; % = 11.25) | | Quit working voluntarily (N = 49; % = 4.96) | | Working more hours (N = 57; % = 5.78) | | |
|---|---|---|---|---|---|---|---|---|---|---|---|
| | N | % | N | % | N | % | N | % | N | % | P-value |
| Gender (Missing = 22) | | | | | | | | | | | 0.1813 |
| Female | 380 | 70.90 | 40 | 7.46 | 58 | 10.82 | 33 | 6.16 | 25 | 4.66 | |
| Male | 307 | 69.46 | 34 | 7.69 | 53 | 11.99 | 16 | 3.62 | 32 | 7.24 | |
| Race & Ethnicity (Missing = 41) | | | | | | | | | | | 0.0005 |
| Non-Hispanic White | 507 | 76.59 | 35 | 5.29 | 59 | 8.91 | 35 | 5.29 | 26 | 3.93 | |
| Hispanic, Latino(a), or Spanish | 54 | 50.94 | 16 | 15.09 | 19 | 17.92 | 4 | 3.77 | 13 | 12.26 | |
| Non-Hispanic Black or African American | 44 | 57.14 | 10 | 12.99 | 15 | 19.48 | 3 | 3.90 | 5 | 6.49 | |
| Non-Hispanic Asian | 16 | 61.54 | 2 | 7.69 | 2 | 7.69 | 2 | 7.69 | 4 | 15.38 | |
| Multiple or other races | 55 | 62.50 | 8 | 9.09 | 14 | 15.91 | 4 | 4.55 | 7 | 7.95 | |
| Age (Missing = 43) | | | | | | | | | | | <0.0001 |
| 18–29 | 49 | 42.98 | 11 | 9.65 | 24 | 21.05 | 9 | 7.89 | 21 | 18.42 | |
| 30–44 | 67 | 55.37 | 18 | 14.88 | 27 | 22.31 | 3 | 2.48 | 6 | 4.96 | |
| 45–64 | 181 | 62.20 | 26 | 8.93 | 46 | 15.81 | 14 | 4.81 | 24 | 8.25 | |
| 65+ | 375 | 87.01 | 17 | 3.94 | 12 | 2.78 | 23 | 5.34 | 4 | 0.93 | |
| Education level (Missing = 26) | | | | | | | | | | | 0.0509 |
| Less than high school diploma | 25 | 75.76 | 3 | 9.09 | 2 | 6.06 | 1 | 3.03 | 2 | 6.06 | |
| Grade 12 or GED (high school graduate) | 145 | 66.82 | 22 | 10.14 | 33 | 15.21 | 8 | 3.69 | 9 | 4.15 | |
| College 1 year to 3 years (some college or technical school) | 241 | 68.08 | 26 | 7.34 | 44 | 12.43 | 19 | 5.37 | 24 | 6.78 | |
| College 4 years or more (college graduate) | 132 | 68.39 | 11 | 5.70 | 25 | 12.95 | 11 | 5.70 | 14 | 7.25 | |
| Graduate or professional degree | 144 | 81.36 | 10 | 5.65 | 6 | 3.39 | 9 | 5.08 | 8 | 4.52 | |
| Income level (Missing = 192) | | | | | | | | | | | 0.0030 |
| $0 - $30,000 | 133 | 58.59 | 21 | 9.25 | 45 | 19.82 | 14 | 6.17 | 14 | 6.17 | |
| $30,001 - $65,000 | 158 | 71.17 | 17 | 7.66 | 27 | 12.16 | 10 | 4.50 | 10 | 4.50 | |
| $65,001 - $105,000 | 136 | 73.51 | 16 | 8.65 | 14 | 7.57 | 7 | 3.78 | 12 | 6.49 | |
| $105,001 or more | 134 | 77.01 | 8 | 4.60 | 12 | 6.90 | 11 | 6.32 | 9 | 5.17 | |
| Residential area (Missing = 13) | | | | | | | | | | | 0.2288 |
| Urban | 581 | 69.41 | 66 | 7.89 | 101 | 12.07 | 42 | 5.02 | 47 | 5.62 | |
| Rural | 115 | 76.67 | 8 | 5.33 | 10 | 6.67 | 7 | 4.67 | 10 | 6.67 | |

had a significant association with employment status change: race/ethnicity (p-value <0.0001), age (p-value <0.0001), and income (p-value = 0.0030). Though not significant at p-value <0.05, education level was near significant to employment status change, with a p-value of 0.0509. Of the subsample of 535 participants who reported working full or part-time before the pandemic, 49.43% did not experience a change, and 20.38% were fired or laid off. The same covariates had a significant association with employment status change, and education became significant. See Appendix Table 1A in S1 File for full results.

Regarding changes in household time spent on housework, Table 3 shows that, of the full sample, 62.45% of participants spent the same amount of time doing housework. However, 5.12% of them spent less time, and another 32.43% spent more time. The significantly associated covariates include gender (p-value = 0.0123), race/ethnicity (p-value <0.0001), age (p-value <0.0001), and residential area (p-value = 0.0051). Of the subsample, 54.97% of participants spent the same amount of time doing housework; 5.44% spent less time, and 39.59%

**Table 3. Frequencies and proportions in covariates by housework time change from a sample (n = 996) of Nevada adults in December 2020.** The p-value was computed by the chi-square test.

| Variable | More time (N = 323; % = 32.43) | | Same time (N = 622; % = 62.45) | | Less time (N = 51; % = 5.12) | | P-value |
|---|---|---|---|---|---|---|---|
| | N | % | N | % | N | % | |
| Gender (Missing = 13) | | | | | | | 0.0123 |
| Female | 189 | 34.74 | 319 | 58.64 | 36 | 6.62 | |
| Male | 134 | 30.25 | 294 | 66.37 | 15 | 3.39 | |
| Race & Ethnicity (Missing = 32) | | | | | | | <0.0001 |
| Non-Hispanic White | 189 | 28.29 | 450 | 67.37 | 29 | 4.34 | |
| Hispanic, Latino(a), or Spanish | 49 | 46.23 | 52 | 49.06 | 5 | 4.72 | |
| Non-Hispanic Black or African American | 32 | 40.51 | 37 | 46.84 | 10 | 12.66 | |
| Non-Hispanic Asian | 12 | 46.15 | 14 | 53.85 | 0 | 0.00 | |
| Multiple or other races | 31 | 34.83 | 54 | 60.67 | 4 | 4.49 | |
| Age (Missing = 34) | | | | | | | <0.0001 |
| 18–29 | 58 | 50.88 | 53 | 46.49 | 3 | 2.63 | |
| 30–44 | 60 | 48.00 | 55 | 44.00 | 10 | 8.00 | |
| 45–64 | 102 | 35.05 | 171 | 58.76 | 18 | 6.19 | |
| 65+ | 93 | 21.33 | 324 | 74.31 | 19 | 4.36 | |
| Education level (Missing = 16) | | | | | | | 0.7353 |
| Less than high school diploma | 15 | 46.88 | 15 | 46.88 | 2 | 6.25 | |
| Grade 12 or GED (high school graduate) | 68 | 31.19 | 138 | 63.30 | 12 | 5.50 | |
| College 1 year to 3 years (some college or technical school) | 112 | 31.02 | 228 | 63.16 | 21 | 5.82 | |
| College 4 years or more (college graduate) | 67 | 34.36 | 119 | 61.03 | 9 | 4.62 | |
| Graduate or professional degree | 56 | 31.46 | 115 | 64.61 | 7 | 3.93 | |
| Income level (Missing = 184) | | | | | | | 0.2725 |
| $0 - $30,000 | 75 | 32.89 | 136 | 59.65 | 17 | 7.46 | |
| $30,001 - $65,000 | 75 | 33.33 | 139 | 61.78 | 11 | 4.89 | |
| $65,001 - $105,000 | 53 | 28.49 | 124 | 66.67 | 9 | 4.84 | |
| $105,001 or more | 62 | 35.03 | 111 | 62.71 | 4 | 2.26 | |
| Residential area (Missing = 4) | | | | | | | 0.0051 |
| Urban | 291 | 34.40 | 511 | 60.40 | 44 | 5.20 | |
| Rural | 32 | 21.33 | 111 | 74.00 | 7 | 4.67 | |

spent more time. Like the full sample, gender (p-value = 0.0165), race/ethnicity (p-value = 0.0286), age (p-value = 0.0002), and residential area (p-value = 0.0444) were significant covariates, and income also became significant (p-value = 0.0213) (see Appendix Table 2a in S1 File). Moreover, among the 323 participants who responded that their households spent more time on housework, 47.68% reported that the respondent ("myself") was doing the additional housework. This was especially true for participants who identified as female, multiple or other races, aged 65 or over, had some college or technical school, had annual household income of less than $30,000, and lived in rural areas (see Table 4). Chi-square analysis revealed that gender (p-value <0.0001) and age (p-value = 0.0138) were significantly associated covariates.

In the multinomial logistic model with the full sample, age (p-value <0.0001) and income (p-value = 0.0024) were significantly associated with employment status change. Table 5 shows that, compared to people aged 18–29, people aged 65 or older had a significantly lower odds of working reduced hours (OR = 0.21; 95% confidence interval [CI] = 0.08, 0.56), being fired or laid off (OR = 0.08; 95% CI = 0.03, 0.19), quitting working voluntarily (OR = 0.29; 95%

**Table 4. Frequencies and proportions for covariates by perception of who was doing the additional household work from a sample (n = 323) of Nevada adults in December 2020.** The p-value was computed by the chi-square test.

| Variable | Myself (N = 154; % = 47.68) | | Not myself (N = 169; % = 52.32) | | |
|---|---|---|---|---|---|
| | N | % | N | % | P-value |
| Gender (Missing = 0) | | | | | <0.0001 |
| Female | 114 | 60.32 | 75 | 39.68 | |
| Male | 40 | 29.85 | 94 | 70.15 | |
| Race & Ethnicity (Missing = 10) | | | | | 0.4573 |
| Non-Hispanic White | 92 | 48.68 | 97 | 51.32 | |
| Hispanic, Latino(a), or Spanish | 21 | 42.86 | 28 | 57.14 | |
| Non-Hispanic Black or African American | 13 | 40.63 | 19 | 59.38 | |
| Non-Hispanic Asian | 5 | 41.67 | 7 | 58.33 | |
| Multiple or other races | 19 | 61.29 | 12 | 38.71 | |
| Age (Missing = 10) | | | | | 0.0138 |
| 18–29 | 21 | 36.21 | 37 | 63.79 | |
| 30–44 | 29 | 48.33 | 31 | 51.67 | |
| 45–64 | 42 | 41.18 | 60 | 58.82 | |
| 65+ | 56 | 60.22 | 37 | 39.78 | |
| Education level (Missing = 5) | | | | | 0.7525 |
| Less than high school diploma | 6 | 40.00 | 9 | 60.00 | |
| Grade 12 or GED (high school graduate) | 30 | 44.12 | 38 | 55.88 | |
| College 1 year to 3 years (some college or technical school) | 59 | 52.68 | 53 | 47.32 | |
| College 4 years or more (college graduate) | 31 | 46.27 | 36 | 53.73 | |
| Graduate or professional degree | 26 | 46.43 | 30 | 53.57 | |
| Income level (Missing = 58) | | | | | 0.1276 |
| $0 - $30,000 | 43 | 57.33 | 32 | 42.67 | |
| $30,001 - $65,000 | 37 | 49.33 | 38 | 50.67 | |
| $65,001 - $105,000 | 22 | 41.51 | 31 | 58.49 | |
| $105,001 or more | 24 | 38.71 | 38 | 61.29 | |
| Residential area (Missing = 0) | | | | | 0.5157 |
| Urban | 137 | 47.08 | 154 | 52.92 | |
| Rural | 17 | 53.13 | 15 | 46.88 | |

CI = 0.10, 0.78), and working more hours (OR = 0.03; 95% CI = 0.01, 0.13). Two other age levels (30–44 and 45–64) had a significantly lower odds of working more hours. The ORs were strictly less than 1 at each income level for fired or laid off. Compared to individuals with a household income of $0-$30,000, the other three income levels had a significantly lower odds of being fired or laid off. Respondents with a household income of $105,001 or more also had a significantly lower odds of reduced hours by 0.29 times (95% CI = 0.11, 0.78). Notably, 182 participants did not provide their annual income. Like the full sample, the multinomial model with the subsample results in age (p-value = 0.0431) and income (p-value <0.0001) being significant. See Appendix Table 3A in S1 File. Appendix Table 4A in S1 File shows the demographic distribution of the full sample and regression sample.

In the proportional model with the full sample, age was the only covariate significantly associated with the change in household-level time spent on housework, with a p-value <0.0001. Table 6 shows that, compared to respondents aged 18–29, the odds of spending less time on housework was significantly higher by 2.14 times (95% CI = 1.28, 3.60) in respondents aged 45–64 and by 3.32 times (95% CI = 1.98, 5.58) in respondents aged 65 or over. Similarly, age

**Table 5. Odds ratios of employment status change for covariates from a sample (n = 766) of Nevada adults in December 2020.**

| Variable | Reduced hours vs. Has not changed | | | Fired or laid off vs. Has not changed | | | Quit working voluntarily vs. Has not changed | | | Working more hours vs. Has not changed | | | |
|---|---|---|---|---|---|---|---|---|---|---|---|---|---|
| | OR | 95% CI | | OR | 95% CI | | OR | 95% CI | | OR | 95% CI | | P-value |
| Gender | | | | | | | | | | | | | 0.5162 |
| Female | 1.19 | 0.67 | 2.13 | 0.87 | 0.54 | 1.42 | 1.61 | 0.82 | 3.16 | 0.82 | 0.42 | 1.59 | |
| Male | Reference | | | Reference | | | Reference | | | Reference | | | |
| Race & Ethnicity | | | | | | | | | | | | | 0.7463 |
| Non-Hispanic White | Reference | | | Reference | | | Reference | | | Reference | | | |
| Hispanic, Latino(a), or Spanish | 1.56 | 0.67 | 3.66 | 1.17 | 0.57 | 2.39 | 0.79 | 0.24 | 2.65 | 2.13 | 0.83 | 5.46 | |
| Non-Hispanic Black or African American | 2.53 | 1.07 | 5.96 | 1.97 | 0.91 | 4.27 | 0.93 | 0.26 | 3.29 | 1.81 | 0.54 | 6.00 | |
| Non-Hispanic Asian | 1.51 | 0.31 | 7.41 | 0.38 | 0.05 | 3.13 | 1.32 | 0.26 | 6.76 | 2.26 | 0.54 | 9.51 | |
| Multiple or other races | 1.14 | 0.41 | 3.21 | 1.40 | 0.65 | 3.02 | 0.79 | 0.22 | 2.77 | 1.83 | 0.65 | 5.14 | |
| Age | | | | | | | | | | | | | <0.0001 |
| 18–29 | Reference | | | Reference | | | Reference | | | Reference | | | |
| 30–44 | 1.24 | 0.48 | 3.22 | 0.94 | 0.43 | 2.03 | 0.23 | 0.05 | 0.98 | 0.31 | 0.10 | 0.90 | |
| 45–64 | 0.72 | 0.29 | 1.78 | 0.60 | 0.30 | 1.22 | 0.35 | 0.12 | 1.03 | 0.36 | 0.15 | 0.84 | |
| 65+ | 0.21 | 0.08 | 0.56 | 0.08 | 0.03 | 0.19 | 0.29 | 0.10 | 0.78 | 0.03 | 0.01 | 0.13 | |
| Education level | | | | | | | | | | | | | 0.8603 |
| Less than high school diploma | Reference | | | Reference | | | Reference | | | Reference | | | |
| Grade 12 or GED (high school graduate) | 0.91 | 0.18 | 4.54 | 0.52 | 0.10 | 2.63 | 0.94 | 0.10 | 8.57 | 1.61 | 0.28 | 9.21 | |
| College 1 year to 3 years (some college or technical school) | 1.11 | 0.54 | 2.29 | 1.23 | 0.67 | 2.26 | 1.85 | 0.70 | 4.95 | 1.85 | 0.74 | 4.66 | |
| College 4 years or more (college graduate) | 0.75 | 0.29 | 1.93 | 1.43 | 0.70 | 2.92 | 2.03 | 0.69 | 5.98 | 1.98 | 0.69 | 5.69 | |
| Graduate or professional degree | 0.98 | 0.38 | 2.52 | 0.52 | 0.19 | 1.41 | 1.62 | 0.51 | 5.16 | 1.22 | 0.35 | 4.30 | |
| Income level | | | | | | | | | | | | | 0.0024 |
| $0 - $30,000 | Reference | | | Reference | | | Reference | | | Reference | | | |
| $30,001 - $65,000 | 0.66 | 0.32 | 1.37 | 0.45 | 0.25 | 0.82 | 0.53 | 0.22 | 1.25 | 0.45 | 0.18 | 1.14 | |
| $65,001 - $105,000 | 0.69 | 0.32 | 1.50 | 0.26 | 0.13 | 0.52 | 0.47 | 0.18 | 1.24 | 0.60 | 0.24 | 1.48 | |
| $105,001 or more | 0.29 | 0.11 | 0.78 | 0.19 | 0.09 | 0.42 | 0.77 | 0.31 | 1.91 | 0.56 | 0.20 | 1.51 | |
| Residential area | | | | | | | | | | | | | 0.6494 |
| Urban | 1.14 | 0.47 | 2.73 | 1.48 | 0.67 | 3.27 | 1.03 | 0.41 | 2.61 | 0.60 | 0.23 | 1.55 | |
| Rural | Reference | | | Reference | | | Reference | | | Reference | | | |

was the only significant variable in the subsample (p-value = 0.0257), with those aged 65 or older being 2.24 times (95% CI = 1.13, 4.43) and those aged 45–64 being 2.17 times (95% CI = 1.22, 3.89) more likely to report the household spent less time on housework than those aged 18–29 (see Appendix Table 5A in S1 File).

In the logistic model examining only those who reported their households did more housework, gender was the only significant covariate for perceptions of primary housework responsibility, with a p-value = 0.0002, where females had a significantly higher odds of perceiving themselves doing the additional housework than males by 2.90 times (95% CI = 1.66, 5.05). See Table 7. For the demographic breakdown of this subsample, see Appendix Table 6A in S1 File.

## Discussion

In 2020, disruptions caused by COVID-19 were ubiquitous. Nevada proves a unique case study given the state's record-high pandemic job losses and the economic context. Our survey of Nevada adults, conducted in December 2020, sought to assess differences in employment status and perceptions of housework responsibilities before large-scale vaccine uptake and

**Table 6. Odds ratios of household level housework time change for covariates from a sample (n = 775) of Nevada adults in December 2020.**

| Variable | OR | 95% CI | | P-value |
|---|---|---|---|---|
| Gender | | | | 0.6117 |
| Female | 0.93 | 0.69 | 1.25 | |
| Male | | Reference | | |
| Race & Ethnicity | | | | 0.8977 |
| Non-Hispanic White | | Reference | | |
| Hispanic, Latino(a), or Spanish | 0.86 | 0.52 | 1.42 | |
| Non-Hispanic Black or African American | 0.92 | 0.54 | 1.56 | |
| Non-Hispanic Asian | 0.66 | 0.27 | 1.64 | |
| Multiple or other races | 1.01 | 0.59 | 1.71 | |
| Age | | | | <0.0001 |
| 18–29 | | Reference | | |
| 30–44 | 1.33 | 0.75 | 2.36 | |
| 45–64 | 2.14 | 1.28 | 3.60 | |
| 65+ | 3.32 | 1.98 | 5.58 | |
| Education level | | | | 0.1401 |
| Less than high school diploma | | Reference | | |
| Grade 12 or GED (high school graduate) | 0.31 | 0.12 | 0.76 | |
| College 1 year to 3 years (some college or technical school) | 0.77 | 0.51 | 1.15 | |
| College 4 years or more (college graduate) | 0.75 | 0.47 | 1.20 | |
| Graduate or professional degree | 0.77 | 0.47 | 1.27 | |
| Income level | | | | 0.3851 |
| $0 - $30,000 | | Reference | | |
| $30,001 - $65,000 | 0.93 | 0.63 | 1.38 | |
| $65,001 - $105,000 | 1.17 | 0.76 | 1.80 | |
| $105,001 or more | 0.79 | 0.51 | 1.24 | |
| Residential area | | | | 0.2525 |
| Urban | 0.78 | 0.51 | 1.20 | |
| Rural | | Reference | | |

during major pandemic-related restrictions and changes in behavior. Overall, we found that about 30% of respondents reported experiencing some change in employment, with about 23% experiencing changes likely to negatively affect household income (i.e., reduced hours, fired, laid off, or quit). This may be indicative of a proportion of Nevadans who experienced changes in resources with implications for household security. Roughly one-third of participants reported that their households spent more time on housework. Female respondents were more likely than male respondents to say that their household spent more time on housework during the pandemic. Gender differences in perceived household time spent on housework, however, did not hold significance in the multivariate analysis, suggesting other factors, such as age or income, may be at play. Of respondents reporting that their households spent more time on housework, females were more likely than males to report themselves doing the additional housework.

Our finding that female employment status change between March and December 2020 was similar to that of males differs from other studies. For example, a study of four other western states, Raile and colleagues [28] found that females were more likely than males to be laid off and lose income and less likely to be designated as essential workers. An analysis of national employment data reported that women with a high school diploma or less were much more likely to leave the labor force between the third quarters of 2019 and 2021 than males with

**Table 7. Odds ratios of perceptions of who did more housework for covariates from a sample (n = 256) of Nevadan adults in December 2020.**

| Variable | OR | 95% CI | | P-value |
|---|---|---|---|---|
| Gender | | | | 0.0002 |
| Female | 2.90 | 1.66 | 5.05 | |
| Male | | Reference | | |
| Race & Ethnicity | | | | 0.7255 |
| Non-Hispanic White | | Reference | | |
| Hispanic, Latino(a), or Spanish | 1.05 | 0.47 | 2.33 | |
| Non-Hispanic Black or African American | 0.59 | 0.24 | 1.47 | |
| Non-Hispanic Asian | 1.33 | 0.33 | 5.41 | |
| Multiple or other races | 1.35 | 0.51 | 3.58 | |
| Age | | | | 0.0737 |
| 18–29 | | Reference | | |
| 30–44 | 1.52 | 0.63 | 3.63 | |
| 45–64 | 1.32 | 0.59 | 3.00 | |
| 65+ | 2.94 | 1.21 | 7.16 | |
| Education level | | | | 0.4002 |
| Less than high school diploma | | Reference | | |
| Grade 12 or GED (high school graduate) | 0.54 | 0.13 | 2.16 | |
| College 1 year to 3 years (some college or technical school) | 1.27 | 0.60 | 2.67 | |
| College 4 years or more (college graduate) | 1.01 | 0.42 | 2.44 | |
| Graduate or professional degree | 0.61 | 0.23 | 1.59 | |
| Income level | | | | 0.2408 |
| $0 - $30,000 | | Reference | | |
| $30,001 - $65,000 | 0.57 | 0.28 | 1.17 | |
| $65,001 - $105,000 | 0.45 | 0.20 | 1.03 | |
| $105,001 or more | 0.55 | 0.25 | 1.25 | |
| Residential area | | | | 0.5714 |
| Urban | 0.77 | 0.31 | 1.90 | |
| Rural | | Reference | | |

similar education [29]. One potential reason for our dissimilar findings may be that our sample exhibited higher educational attainment levels compared to the overall Nevada population; only about 25% of our sample had a high school diploma or less education compared to about 41% in the state overall [30]. This may have obfuscated some of the gender employment disparities. Another factor may be that males and females are equally represented in the tourism, gaming, and entertainment industry, which constitutes a large proportion of the state's economy [31]. However, previous studies reported that women were more likely than men to lose their jobs during the pandemic, even after controlling for job characteristics, such as industry, occupation, and the ability to work from home [32, 33]. Further examination is necessary to better understand the underlying causes of the differences in our sample.

The proportion of respondents who reported that their employment status did not change since the start of the pandemic increased with age, with 87% of those aged 65 or older reporting no change. After controlling for sociodemographic variables in the regression model, those aged 65 or older remained significantly less likely to report employment change. This finding is not surprising, given that the largest proportion of retirees and individuals out of the labor market are in this age bracket [34]. This is also evidenced by our subsample analysis of only those working prior to the start of the pandemic. The largest proportion of individuals who

were not working were at least 65. Further, in this subsample, the proportion of those who reported no change in this age group became much more like other age groups. The finding that those aged 65 or older were significantly more likely to report that they quit working voluntarily is similar to work by the Federal Reserve Bank of St. Louis, which estimates that about three million people retired early due to COVID-19 and speculates that "a significant number of people who had not planned to retire in 2020 may have retired anyway because of the dangers to their health or due to rising asset values that made retirement feasible" [35]. Those aged 30–64 were significantly less likely to work more hours. While more research is needed to fully understand this trend, it is possible that those in that age bracket have greater job tenure and thus greater job stability [36, 37].

Although job settings can be sources of COVID-19 exposure, employment is also a critical health determinant. It impacts health through, for example, income and resource availability, access to work-based benefits like health insurance, and networks of social support [38]. Therefore, racial and ethnic disparities in employment loss may have far-reaching impacts and enhancing equity in employment is critical for health outcomes down the line. This analysis suggests that employment changes differed by race/ethnicity in Nevada during this pandemic phase. The pandemic contributed to a sizable reduction in overall work hours, layoffs, and firings for Nevadans, with disproportionate impacts in communities of color, particularly among Black, Hispanic/Latinx, and multiracial/other individuals. This finding is not surprising and is consistent with other studies that find differences in employment by race/ethnicity. For instance, U.S. Census Current Population Survey (CPS) data analyses revealed that (1) Black, Hispanic, and Asian American groups experienced greater employment declines compared to Whites and non-Hispanics between the first and second quarters of 2020 [39], (2) compared to their White counterparts, Hispanic and Black individuals were at an increased risk of employment-related income loss early in the pandemic [40], and (3) differences between unemployment rates for Whites–compared to both Blacks and Hispanics–grew between April and June 2020 and were contributed to by a lower likelihood among Blacks and Hispanics to hold jobs that could be performed from home [24]. Others have indicated that women and Hispanic, Black, and Native American workers make up larger percentages of sectors most directly impacted by pandemic-related closures and distancing requirements, including travel, transportation, service, entertainment, and retail [41].

Similar reasoning–that Blacks/African Americans and Hispanics/Latinx workers were less likely to hold positions easy to translate to a "work from home" world–is consistent with Nevada's pre-COVID workforce demographics, where these groups are disproportionately represented in the food establishment, retail, and hospitality industries [22]. It may help explain some of this pattern. Our findings that the proportion of people with no change in employment increased with income and the proportion of those fired decreased with income may also be explained, at least in part, by employment sector demographics and related pay. For example, in the Las Vegas metropolitan area, which constitutes over 70% of Nevada's population, food preparation and service jobs make up 12.7% of total employment (compared to 8.1% for the U.S.), with a mean hourly wage of $13.59 and grounds cleaning and maintenance jobs make up 4.9% of total employment (compared to 2.9% for the U.S.), with a mean hourly wage of $15.96 [42–44]. In contrast, higher-earning occupational group sectors (e.g., business, computer and mathematical, legal, and educational instruction) are under-represented in Las Vegas compared to the nation [44]. These higher-paying jobs may have translated better to a work-from-home landscape, resulting in their ability to weather the pandemic [45]. This explanation is bolstered by our significant finding for education in the subsample (and near significant finding in the full sample); the most educated were most likely to report that their

jobs had not changed and least likely to be fired and laid off, though these findings were ultimately not significant in the model.

These pre- and post-pandemic employment patterns are also consistent with pre-existing patterns of discrimination in employment. For instance, Black/African American workers are often the first laid off during a downturn [46]. Our findings are consistent with both explanations, which may, in fact, be interrelated. It is not surprising that race did not remain significant in the model. Recent equity research focused on intersectionality—the convergence of several identity variables tied to advantage and disadvantage—may better explain disparities than considering individual variables in isolation [47]. While we did attempt to examine intersectionality, the sample size for numerous categories grew too small to draw meaningful conclusions. Future studies may consider oversampling populations of interest.

The significance of employment as a health determinant and the link between loss of work and health outcomes make the racial/ethnic and income disparities evident in our analysis concerning. Research suggests that job loss related to COVID-19 is linked to health outcomes. Studies indicate that those who experienced COVID-19-related job loss exhibited more stress, anxiety, and depression symptoms and less positive mental health than those who did not experience such job losses and that losing a job since the pandemic began was associated with having more unhealthy mental health days, while working reduced hours due to the pandemic was associated with more unhealthy mental and physical health days [48, 49]. Detrimental changes in employment may be yet another avenue perpetuating or exacerbating health disparities based on race/ethnicity and income far beyond the pandemic.

Between one-third (full sample) and 40% (subsample) of respondents indicated that their household spent more time on housework compared to before the pandemic. This presents a serious challenge, especially when coupled with disruptions in employment and income catalyzed by the pandemic. Our finding that the proportion of those who reported more time doing housework during the pandemic decreased with age, which was significant in the proportional model(s), may be connected to participants' definitions of housework. If they included childcare in the definition, younger participants, who are more likely parents caring for younger children, could be expressing housework changes stemming from closed or remote schools and childcare facilities. This interpretation is consistent with findings that parents' childcare responsibilities increased during the pandemic [50]. Our finding that those aged 65 and older were least likely to report changes in housework is also consistent with the interpretation that childcare may be a driver of this perceived change.

Responses also revealed the elevated toll of the pandemic on females when it comes to housework. Females were more likely to report that their household experienced increased housework. This finding may be explained by factors other than gender, like income and education, because it did not hold in the model. Nevertheless, this perceived change in household-level housework, more likely to be articulated by females, is meaningful. When examining only respondents who reported a household-level increase in housework, females were more likely than males to report that they were the ones actually doing the additional work. A recent review by Yavorsky and colleagues [50] suggests that pre-COVID gender inequities in household responsibilities may have worsened: In the pandemic, women conducted more housework and provided more childcare than men (although findings are mixed on the overall changes in household labor by parents), experienced distress connected to working and providing childcare simultaneously, and decreased engagement in the workforce to attend to household and childcare responsibilities. An intersectional perspective suggests that the perceptions of increased housework may have been even starker had the education, income, race/ethnicity, and other variables of this group of respondents put them at a greater relative disadvantage.

This study should be viewed in light of its limitations. As a cross-sectional study, the temporal relationship between the independent and dependent variables often cannot be established, limiting the identification of causal relationships. The sample was one of convenience, so those who responded may differ in some way from those who did not. Respondents self-reported their perceptions of the variables of interest, which was appropriate for our study purposes, but self-reported data are subject to bias. The generalizability of findings may be limited due to demographic differences between the sample population and the state of Nevada. Most notably, compared to the state, the sample population had a much higher proportion of respondents who were aged 65 or older and a much lower proportion of respondents under 45 years; a higher percentage of respondents who were White, female, and held a graduate or professional degree; and a lower percentage of Hispanics, Asians, and those reporting "other" or multiple races. Respondents were limited to Nevada and completed the survey in English, which may limit linguistic, economic, and geographic generalizability. The sample size for the subgroup analysis was smaller and power was not calculated. Finally, as the spread of COVID-19 and the community response changed over time, perceptions, attitudes, and behaviors may have changed as well. While survey questions asked about the pandemic in general, respondents' perceptions may have been influenced by recent events and not truly reflect this longer time period. Relatedly, pandemic fatigue and pandemic burnout due to an unprecedented pandemic and associated restrictions may have played a role in participation and general perceptions and behaviors at the time of the survey [55, 56]. The survey was conducted in late 2020, at the peak of the alpha variant surge in Nevada, which may have biased the results to behaviors and perceptions closer to the end of 2020 than to the start of the pandemic.

## Conclusions

Changes in employment and housework represent two major stressors stemming from the COVID-19 pandemic. Our study examined how employment status and perceptions of housework changed in a sample of Nevadan adults in December 2020. Its findings confirm age, income, and racial/ethnic disparities in employment changes, highlighting systemic vulnerabilities known to have negative health consequences. We also confirmed age and gender disparities in perceptions of added housework. Nevada's lack of economic diversity and its high racial and ethnic diversity make our findings particularly pertinent, as both factors led to increased population-level vulnerability to the pandemic. Our findings can aid in planning for and mitigating efforts during future emergencies, including pandemics, in Nevada and similarly impacted localities and regions.

## Supporting information

**S1 File.**
(DOCX)

## Author Contributions

**Conceptualization:** Courtney Coughenour, Brian Labus, Maxim Gakh, Pashtana Usufzy.

**Formal analysis:** Lung-Chang Chien.

**Funding acquisition:** Brian Labus.

**Methodology:** Courtney Coughenour, Brian Labus, Maxim Gakh.

**Project administration:** Courtney Coughenour, Brian Labus, Maxim Gakh.

**Writing – original draft:** Courtney Coughenour, Lung-Chang Chien, Brian Labus, Maxim Gakh, Pashtana Usufzy.

**Writing – review & editing:** Courtney Coughenour, Lung-Chang Chien, Brian Labus, Maxim Gakh, Pashtana Usufzy.

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
