## [Decision Letter · Decision Letter 0]

12 Nov 2023

PONE-D-23-10602Shifting employment and perceptions of household responsibilities during early stages of the COVID-19 pandemic in NevadaPLOS ONE

Dear Dr. Coughenour,

Thank you for submitting your manuscript to PLOS ONE. After careful consideration, we feel that it has merit but does not fully meet PLOS ONE’s publication criteria as it currently stands. Therefore, we invite you to submit a revised version of the manuscript that addresses the points raised during the review process. Both reviewers have addressed the merits of the paper. In the meantime, they also raised detailed section-by-section comments on how to enhance the paper quality. The authors are thus requested to revise their paper thoroughly and make sure to address those detailed comments made by the two reviewers, especially those critical comments on methodology and data samples.

We look forward to receiving your revised manuscript.

Kind regards,

Chenfeng Xiong

Academic Editor

PLOS ONE

Journal Requirements:

2.  You indicated that ethical approval was not necessary for your study. We understand that the framework for ethical oversight requirements for studies of this type may differ depending on the setting and we would appreciate some further clarification regarding your research. Could you please provide further details on why your study is exempt from the need for approval and confirmation from your institutional review board or research ethics committee (e.g., in the form of a letter or email correspondence) that ethics review was not necessary for this study? Please include a copy of the correspondence as an ""Other"" file.

3.  In the ethics statement in the Methods, you have specified that verbal consent was obtained. Please provide additional details regarding how this consent was documented and witnessed, and state whether this was approved by the IRB .

“This project was partially supported by the U.S. Centers for Disease Control and Prevention as part of a subaward totaling $3.4 million from the Nevada Division of Public and Behavioral Health's Epidemiology and Laboratory Capacity award. The contents are those of the author(s) and do not necessarily represent the official views of, nor an endorsement by, the U.S. Centers for Disease Control and Prevention or the U.S. government.

Reviewers' comments:

Reviewer's Responses to Questions

**Comments to the Author**

1. Is the manuscript technically sound, and do the data support the conclusions?

Reviewer #1: Yes

Reviewer #2: Partly

2. Has the statistical analysis been performed appropriately and rigorously? 

Reviewer #1: Yes

Reviewer #2: Yes

3. Have the authors made all data underlying the findings in their manuscript fully available?

Reviewer #1: No

Reviewer #2: No

4. Is the manuscript presented in an intelligible fashion and written in standard English?

Reviewer #1: Yes

Reviewer #2: Yes

5. Review Comments to the Author

Reviewer #1: Thank you for the opportunity to read this very interesting paper. The paper has many strengths that should be of interest to the journal audience. Thus, the following suggestions are around enhancing the presentation for publication and clarifying aspects of the data and reporting.

I will go by line number for the most part. If not, I will try to be as specific as possible in noting the area I am speaking about.

Overall

The manuscript should be shortened

Abstract

[Line 47] Background:. It is necessary to include the country.

Where are values and p-values? (p < .001). Authors must specify it. P values showing the differences between groups should be given.

Methods. Did you use some scales? If not, it is necessary to detailed the variables

1. Introduction

Authors must speak more about gender, status, types of works, and race, the adverse working conditions related to COVID-19, which are the consequences of shifting employment. What is the meaning of household responsibilities?

Please consider stating clear in your text which was the exact understudied population (population of interest) and how it was defined.

The prevalence and incidence of COVID-19 in the area of study during the period of study should be discussed.

It is not clear why the study was necessary

Authors must finish with the main aim.

2. Materials and Methods

Did the authors calculate the needed sample size? Please, clarify. How was the sample size determined? Did the authors test power calculation? How was the sample chosen? Authors must specify it.

Do the authors have a study protocol? The study protocol should be described in detail.

Which is the ID number? (ID number…..:2020). Interventionary studies involving animals or humans, and other studies require ethical approval must list the authority that provided approval and the corresponding ethical approval code. Please include the date and code register number of ethics committee.

Please add the response options for each demographic variable in the study

There isn’t enough detail to repeat the experiments.

Are any potential confounding factors considered?

DESIGN AND PROCEDURE: they should specify the design of the study they have carried out, and describe thoroughly how the data collection process was carried out, as well as issues such as voluntariness of participation and/or anonymity.

POPULATION and SAMPLE: It is necessary to describe the population size of the Nevada, and from this data, provide a calculation of the sample size necessary for the results to be meaningful. It is also necessary to specify the inclusion and exclusion criteria for the study sample.

VARIABLES: In relation to the items that were created "ad hoc", it is also necessary to better describe how these items were agreed (literature review, expert consensus, etc.).

ETHICAL CONSIDERATIONS: You should include a sub-heading under Methods that describes these issues: provide the reference number of the Ethics Committee approval, describe how the confidentiality of the data has been guaranteed.

3. Results

Table 1. Please, provide the n and not only %

At last, but not least, I recommend you to make available your data in an open repository. I think it will make this scientific process more transparent, and it allows other researchers to replicate your results.

4. Discussion

I think that the discussion section could be shortened by not repeating survey results

Moreover, some points were not discussed, i.e., the participants were assessed from December, 2020, since a fatigue scenario could exist due to Covid-19 social restrictions; how could this factor impact these participants?

Limitations related with the type of methodology used. Limitations regarding representativeness of respondents should be better addressed. Authors must specify it. The fact of having a convenience sample should be included in the limitations of the study.

I wish you all the best.

Reviewer #2: Intro: First paragraph- would be helpful to have death estimates at time of survey instead of July 2022

Intro: lines 91-92, it is unclear at time of survey what type of closures were in place for gaming industry

Methods- more details are needed about the sampling frame (list of phone numbers) and sampling procedures. Who provided this frame? Were quotas employed for age-sex? etc.

Methods: Please provide a statistical justification for the sample size

Methods: The response rate is a result and not a method. More details about the disposition codes and survey rates are needed. See AAPOR standard guidelines.

Results: Please be consistent with decimal point reporting (i.e sometimes you report 67%, 67.0% and 67.00%

Results: line 185- reword "moderately significant" to something else like, "Although not significant...list pvalue

Results: Table 1 shows demographics of full sample (n=1000) but sample size of 777 is used for main analyeses (Table 5). Please include a supplemental table that looks at demographic distribution of the full sample and regression sample to see if there is any potential selection bias.

Results/Methods: The authors mentioned restricting data set to those without missing data. Income level has the largest amount of missing data (n=192). Did the authors look at relaxing this criteria so that a larger dataset could be used for analyses

Result Table 4...there is an error in percentages for males (29.85% is in both columns)

Results table 7. I would defer to a proper statistician, but it seems like this could be more robustly assessed through an interaction term. If you are presenting Table 7 as a main finding, please include the demographics of this sub-sample.

Results: why not present weighted estimates for some of the key outcomes? especially given the biases the authors mentioned in the demographics of the sample.

Abstract- the reporting that women have more household work is misleading. Overall, there was minimal difference in lesss/more/no diff by gender

Limitations: Need to add that study was (probably, I'm guessing) not powered for sub-group analyses

Results and Abstract... overall I think the results section could be tightened up a little bit. It seems like there is a lot of mention in covariates that are significant in univariate models but not in multivariate models. Table 5 and 6 are the main adjusted regression tables where only age was significantly associated for both outcomes. Quite surprisingly, gender and race/ethnicity are not significantly associated. This gets a little lost in the presentation of results and the discussion. For example, the paragraph starting at line 342. "Responses also revealed the elevated toll of the pandemic on females when it comes to housework?. Consider tempering the conclusion

6. PLOS authors have the option to publish the peer review history of their article (what does this mean?). If published, this will include your full peer review and any attached files.

Reviewer #1: **Yes: **Juan Jesús García-Iglesias

Reviewer #2: No

---

## [Author Response · Author response to Decision Letter 0]

2 Feb 2024

We uploaded a point by point document.

---

## [Decision Letter · Decision Letter 1]

29 Apr 2024

PONE-D-23-10602R1Shifting employment and perceptions of household responsibilities during early stages of the COVID-19 pandemic in Nevada , USAPLOS ONE

Dear Dr. Coughenour,

Thank you for submitting your manuscript to PLOS ONE. After careful consideration, we feel that it has merit but does not fully meet PLOS ONE’s publication criteria as it currently stands. Therefore, we invite you to submit a revised version of the manuscript that addresses the points raised during the review process. Please address the remaining comments from the reviewers.

We look forward to receiving your revised manuscript.

Kind regards,

Chenfeng Xiong

Academic Editor

PLOS ONE

Reviewers' comments:

Reviewer's Responses to Questions

**Comments to the Author**

1. If the authors have adequately addressed your comments raised in a previous round of review and you feel that this manuscript is now acceptable for publication, you may indicate that here to bypass the “Comments to the Author” section, enter your conflict of interest statement in the “Confidential to Editor” section, and submit your "Accept" recommendation.

Reviewer #1: All comments have been addressed

Reviewer #3: (No Response)

2. Is the manuscript technically sound, and do the data support the conclusions?

Reviewer #1: Yes

Reviewer #3: Partly

3. Has the statistical analysis been performed appropriately and rigorously? 

Reviewer #1: Yes

Reviewer #3: No

4. Have the authors made all data underlying the findings in their manuscript fully available?

Reviewer #1: No

Reviewer #3: (No Response)

5. Is the manuscript presented in an intelligible fashion and written in standard English?

Reviewer #1: Yes

Reviewer #3: (No Response)

6. Review Comments to the Author

Reviewer #1: (No Response)

Reviewer #3: (No Response)

7. PLOS authors have the option to publish the peer review history of their article (what does this mean?). If published, this will include your full peer review and any attached files.

Reviewer #1: **Yes: **Juan Jesús García-Iglesias

Reviewer #3: **Yes: **Harumitsu Suzuki

---

## [Author Response · Author response to Decision Letter 1]

12 Jul 2024

1) I understand that the survey was conducted by the firm, but who were the subjects registered with this firm? For example, supermarket users or online shoppers?

Response: Thank you for your inquiry. For cell phone data, the survey firm utilized the Telcordia Terminating Point Master (TPM) data, which is a proprietary database. This database includes detailed information about telephone number blocks within the North American Numbering Plan (NANP).

To clarify, the TPM data is not a list of survey subjects such as supermarket users or online shoppers. Instead, it is a technical resource that telecommunications companies, regulatory bodies, and other related entities use for managing and operating telephone networks. The data includes records for area codes and exchanges, broken down into 1000-number and 100-number blocks.

In the context of our survey, the TPM data was used to ensure comprehensive and representative coverage of telephone numbers within Nevada. This methodology helps in achieving a broad and unbiased sample of telephone users across different geographic regions, rather than targeting specific types of consumers like supermarket users or online shoppers.

For landlines – the firm utilized Directory-Listed Landline telephone samples. 

The following has been added to the results: data collection and survey section: Cellphone lines came from a proprietary data source with comprehensive coverage in Nevada and landlines from directory-listed numbers. Of these sources, phone numbers from individuals residing in Nevada were randomly selected and dialed.

2) Nowhere did it say how to select the subject. Normally, if there was a list of contacts, there would be some way to randomly select participants from that list, but I wonder if such a method was used.

Response: From the participant information from cellphone and landlines listed above, participants that were located in Nevada were randomly selected and dialed by the survey firm. 

3) Is there ever more than one participant from a single household in this survey?　 If more than one person is participating, it is better to make an adjustment for each household. This is because if the husband responds that he spends more time at home and thus does more housework, the wife may respond that she does less housework, and their answers would be influenced by the same household.

Response: No, only one participant per household participated in the survey. We added this to the results section.

4) Why are there so many missings when you are communicating directly by phone?

Response: The “missing” is a combination of “don’t know” and the participant choosing not to answer the question, as participants had the right to choose not to answer. However, only 32% of the “missing” data are the “don’t know” response.

5) I understand that there is missing in household income, but if the respondent had indicated that he/she did not want to answer the question, you could make a no answer category. It would reduce the percentage of missing. Also, you asked about household income, but it is possible that young people, such as 18 years old, may have answered that they did not know. Was there any trend when looking at those who categorized missing by age?

Response: We thank the reviewer for their feedback. The majority of the missing data are responses where participants refused to answer. Upon examining the "don't know" responses by age, we found that only 10 individuals (22%) in the younger age category responded this way. Additionally, when analyzing all income categories within this age group, no apparent trend was observed.

2. For the chi-squared test, all of the expected counts be at least 5 in each cell. However, several categories have the expected counts was below 5, for example, the expected counts of non-Hispanic Asians who responded "Reduced hour" in the Race and Ethnicity category was less than 5. The analysis method is not appropriate. Authors should calculate the p-value using the fisher exact test, etc. If you have a large number of categories, some with very small expected numbers, alternatively, you should consider to summarize categories that are few in number for chi square test. 

Response: We thank the reviewer for this suggestion. Cell frequencies less than 5 do not violate the assumption of the chi-square test, rather “expected frequencies” should not be less than 5. Cell frequencies less than 5 may have an expected frequency larger than 5, depending on corresponding marginal frequencies and the total frequency. The only variable that violated this assumption is the cross-table between race and employment status change in Table 2. Because Fisher's exact test does not work in such a 5 by 5 table, we computed a simulated p-value in Fisher's exact test in R, resulting in a p-value of 0.0005. The simulation is based on the following paper: An Efficient Method of Generating Random R × C Tables with Given Row and Column Totals - Patefield - 1981 - Journal of the Royal Statistical Society: Series C (Applied Statistics) - Wiley Online Library

We updated the p-value in table 2. This does not change overall findings.

3. The author responded that “Univariate analysis did reveal a gender disparity, which lost significance once the covariates were added to the model.”

As for the analysis method, you say it is the univariate model, but I think it will be an unadjusted model. It is not adjusted for confounding at all, so it would be better to check it in several ways. For example, if there is a difference in the association with outcomes by gender, you could introduce an interaction term into the model, and if that shows a significant difference, you could stratify by gender and look at the association with the other covariate.

The same can be done for age. Since there are many age groups, you can add one covariate at a time to the same model to see which variable attenuates the odds ratio, and thereby see the effect of the variable. It is believed that presenting such a method would further lead to better discussions and conclusions

Response: Thank you for your valuable feedback. We did not actually build any univariate models and have replaced the term “univariate” in the abstract with “chi-square”. Table 2 consists of proportions and frequencies tested for significance with the chisquare test. 

We appreciate your suggestions regarding the analysis methods. While the proposed approaches are indeed insightful and could provide additional depth, they fall outside the scope of our current research question. Our analysis, as submitted, adequately addresses the posed research question. We recognize that exploring these additional methods could constitute a comprehensive analysis deserving of its own dedicated study. We will consider these suggestions for future research endeavors.

---

## [Decision Letter · Decision Letter 2]

21 Aug 2024

Shifting employment and perceptions of household responsibilities during early stages of the COVID-19 pandemic  in Nevada , USA

PONE-D-23-10602R2

Dear Dr. Coughenour,

We’re pleased to inform you that your manuscript has been judged scientifically suitable for publication and will be formally accepted for publication once it meets all outstanding technical requirements.

Kind regards,

Chenfeng Xiong

Academic Editor

PLOS ONE

Additional Editor Comments (optional):

Reviewers' comments:

Reviewer's Responses to Questions

**Comments to the Author**

1. If the authors have adequately addressed your comments raised in a previous round of review and you feel that this manuscript is now acceptable for publication, you may indicate that here to bypass the “Comments to the Author” section, enter your conflict of interest statement in the “Confidential to Editor” section, and submit your "Accept" recommendation.

Reviewer #3: All comments have been addressed

2. Is the manuscript technically sound, and do the data support the conclusions?

Reviewer #3: Yes

3. Has the statistical analysis been performed appropriately and rigorously? 

Reviewer #3: Yes

4. Have the authors made all data underlying the findings in their manuscript fully available?

Reviewer #3: Yes

5. Is the manuscript presented in an intelligible fashion and written in standard English?

Reviewer #3: Yes

6. Review Comments to the Author

Reviewer #3: Thank you for giving me the opportunity to read your interesting research. I look forward to reading further reports. My question is solved from your response.　I have no further questions.

7. PLOS authors have the option to publish the peer review history of their article (what does this mean?). If published, this will include your full peer review and any attached files.

Reviewer #3: **Yes: **Harumitsu Suzuki

---

## [Editor Report · Acceptance letter]

3 Oct 2024

PONE-D-23-10602R2 

PLOS ONE

Dear Dr. Coughenour, 

I'm pleased to inform you that your manuscript has been deemed suitable for publication in PLOS ONE. Congratulations! Your manuscript is now being handed over to our production team.

Kind regards, 

on behalf of

Dr. Chenfeng Xiong 

Academic Editor

PLOS ONE